# Face, content, criterion and construct validity assessment of a newly developed tool to assess and classify work–related stress (TAWS– 16)

**Runalika Roy**[1⊙], **Gautham Melur Sukumar**[1⊙*], **Mariamma Philip**[2‡], **Gururaj Gopalakrishna**[1‡]

1 Department of Epidemiology, National Institute of Mental Health and Neuro Sciences (NIMHANS), Bengaluru, Karnataka, India, 2 Department of Biostatistics, National Institute of Mental Health and Neuro Sciences (NIMHANS), Bengaluru, Karnataka, India

⊙ These authors contributed equally to this work.
‡ MP and GG also contributed equally to this work.
* drgauthamnimhans@gmail.com

**Data Availability Statement:** All relevant data are within the paper and its Supporting Information files.

## Abstract

### Introduction

As work-stress, is associated with Non Communicable Diseases, and decreased work productivity, health and economic benefits are expected from periodic work-stress screening among employees using valid and reliable tools. Tool to Assess and classify Work Stress (TAWS– 16) was developed to overcome limitations in existing work-stress assessment tools in India. This study aims to test face, content, criterion and construct validity of TAWS–16 in a sample of managerial-supervisory employees.

### Methods

Nine domain experts rated face and content validity of TAWS– 16. Content validity was measured by Content Validity Indices (I-CVI, S-CVI) and Modified Kappa statistics. Empirical validity was tested by analysing data reported from 356 Information Technology (IT) professionals wherein Exploratory Factor Analysis was conducted for the assessment of Construct Validity. Self-reported data was collected in an unlinked and anonymous manner using a web-link, which was emailed to the study subjects, after initial introductory telephone or personal conversation. Criterion Validity was tested against stress sub-scale of DASS– 21. This study was approved by NIMHANS ethics committee.

### Results

Findings revealed that TAWS– 16 has good face validity. The content validity is acceptable (CVI = 0.829). Construct Validity is appropriate as 60.8% of the total variance was explained by the factors identified in our study. Criterion Validity was moderate (Kappa Value 0.208) due to inappropriate work-stress instrument for comparison with TAWS– 16.

**Funding:** The author(s) received no specific funding for this work.

**Competing interests:** The authors have declared that no competing interests exist.

## Conclusions

Overall, TAWS– 16 demonstrated good face, content and construct validity. It measures work-stressors, coping abilities and psycho-somatic symptoms associated with work-stress. We recommend use of TAWS– 16 for periodic screening and classification of work-stress among employees.

## Introduction

Stress in harmful proportions poses a risk for Non Communicable Diseases (NCDs), mental morbidity, workplace injuries and their sequelae, resulting in increased health care costs and decreased work productivity [1–12]. Work-related stress is the response people may have when presented with work demands and pressures (work stressors) that are not matched to their knowledge and abilities and which challenge their ability to cope (WHO) [13]. Low job security, excessive work demand, lack of job control, monotonous work, low organisational support, adverse physical working conditions, strained inter-personal relationships at work, role conflict, role ambiguity and work-life imbalance are commonly observed work stressors [4, 14]. Employees exposed to such chronic work-stressors are also likely to develop NCDs over time.

Census reports estimate that there are nearly 530 million workers in India [15] of which 8–9% are in organized sector [16]. Among the organized workplaces, service sector (Information Technology, Business Transformation [BT], Business Process Outsourcing [BPO] and others.) is rapidly growing and has relatively younger workforce, working in globally competitive and stressful work environment. NCDs and NCD risk factors are emerging as leading health priorities among these employees. The COVID– 19 pandemic has added newer work dimensions and its associated risk factors.

Available studies indicate that work-stress among IT professionals ranged between 5–51% [1] with increased prevalence among employees in managerial role (55%) [6]. However, these studies differed in stress assessment tools, sample size, worker profile, geographical location and time-periods. Lack of standard instruments to specifically assess work-related stressors is a key limitation in these studies.

In occupational health practice, regular and periodic screening for work-related stress is limited due to lack of tools specific to measure work–related stressors and stress. Commonly used tools capture stress-in-general and are not specific to work-stressors, are too lengthy and not validated for Indian workplaces. These tools do not account for coping abilities of employees or consider their experience of psychosomatic symptoms, to identify and classify work-stress experience of employees. Copyright and costs of study instruments from high income countries also limit application in low resource settings and in research [17].

With stress and its concomitant health and economic issues expected to rise in India, clearly there is an unmet need for a brief, valid, reliable and easy-to-use tool to identify and categorise work-stress. Towards this direction, a work-stress assessment tool (Tool to assess and classify work-stress i.e., TAWS– 16) [18] has been developed to overcome limitations of existing tools. This study presents the face, content, criterion and construct validity of TAWS– 16.

## Materials and methods

This validation study was conducted on a convenient sample of 356 managerial and supervisory level employees from select information technology companies in Bengaluru city between

September 2020 to March 2021. Employees with minimum 6 months of employment in their present company were included. TAWS– 16 consists of 32 items (16 items across 2 sections). As a thumb rule, a minimum of 320 employees were studied for assessment of empirical validity (10 subjects per item) [19] and subsequently sampling adequacy was tested during analysis by KMO co-efficient (0.843) [20]. Face and Content validity were assessed on a sample of 9 domain experts. Ethical clearance was obtained from Institutional Ethics committee, NIM-HANS (NO.NIMH/DO/IEC (BS & NS DIV)2020–21) and informed consent (verbal and through web-link) was obtained from each study participant.

## Study instrument

Tool to assess and classify work-stress (TAWS– 16) was developed at Department of Epidemiology, Centre for Public Health, National Institute of Mental Health and Neurosciences (NIMHANS). TAWS– 16 comprises of 2 sections (A and B) having 16 items each. Items in Section-A enquire regarding work-stressor experience of employees over the past 6 months of employment and categorises their work-stress levels.

Domains of role in the organisation, role overload, role ambiguity (3 items), career development, effort-reward imbalance and job security (4 items), organisational environment, working conditions, job control, job demand, relationship with peers (5 items), organisational support (3 items) and work life balance (1 item) are covered in Section A. Details regarding items., responses, scoring for each item and cut-off scores for categorisation of work-stress levels is available in the Instruction Manual of TAWS-16 [18].

Employees affirmatively reporting work-stressor experience in last 6 months (for each item in Section A), are further enquired regarding their ability to cope or efficiently manage each reported work stressor. Subjects scoring >48 in Section A, are classified as having work-stress with different cut-off scores for mild, moderate and severe work-stress.

Example: *Item 1*: *I experience time/deadline pressures due to heavy workload*. Employee is expected to provide one response (Yes, to a great extent = 2; Yes to some extent = 1, No, not at all = 0). Employees responding 1 or 2 are further asked *"If they feel stressed or distressed by the stressor?"* to which there are four responses ("No, not at all = 1; Yes, on few occasions, but I manage = 2, Yes, often and difficult to manage = 3, Yes, very frequently and excessively stressed, difficult to manage = 4). There is an objective criteria to operationally define each of these 4 responses.

Items in Section-B enquire regarding commonly reported psychosomatic symptoms suggestive of work-stress. Employees experiencing symptoms are further enquired into frequency of symptoms.

Based on scores obtained in Section-B, employees are classified into different symptom experience categories (mild, moderate and severe).

A cross-tabulation of different categories of work-stress levels and symptom levels is used to provide a colour code and intervention guide for occupational health managers, from an NCD risk reduction perspective.

## Data collection

**Assessment of face validity.** Face validity assess whether each item represents the construct being studied, based on its face value [21]. Nine experts (Public Health Experts/Research Methodologists, Occupational Health Experts, Psychologists, Psychiatrists, IT professional) were identified and informed consent was obtained. The experts were provided a hard copy of TAWS-16 and they rated face validity of each item (on a scale of 10) in terms of its understanding ability, unambiguousness, clarity and chance of misinterpretation. Average scores were

calculated for rating of understanding ability, unambiguousness, clarity, chance of misinterpretation and overall face validity.

**Assessment of content validity.** Content validity is the degree to which the instrument covers all items necessary or sufficient to measure the construct of interest [22]. Each item was reviewed by 8 experts who rated content validity of each item in terms of its relevance to assess work-stress. Experts identified were different but similar in profile to experts who rated the face validity. Content validity rating for each item consisted of a "4-response" scale (Must be required = 3, Required = 3, Useful, but not required = 1, Not useful, must be removed = 0).

Content Validity Indices namely Item level Content Validity Index (I–CVI), Scale level Content Validity Index (S–CVI) [23–25] and Multi–rater Kappa Statistic (Modified Kappa) were calculated [26] to measure content validity. For calculation of I–CVI and S–CVI, the response categories for each item were dichotomized by combining responses 3 and 2 together as "relevant" and 1 and 0 together as "not relevant". Details of calculation and criterion as provided in Table 1.

Kappa statistic is a measure of inter–rater agreement that adjusts for chance agreement and is an important supplement to CVI because Kappa provides information about degree of agreement beyond chance [25, 29]. Modified Kappa statistic (K) was computed and agreement was categorised based on the K values [28].

**Assessment of criterion validity.** Criterion validity is assessed to determine the relationship of scores on a test to a specific criterion [30, 31]. As there was no publicly available standard questionnaire specific to assess and classify work-related stress for Indian settings, TAWS– 16 was compared to the most commonly used stress tool. Based on review of literature and interaction with occupational health experts, criterion validity was assessed against the commonly used DASS– 21: Depression, Anxiety, Stress Scale– 21 [32] even though it was not specific to work–related stress. TAWS– 16 was compared against the stress sub-scale of DASS– 21 which consists of 7 items [32]. Criterion (concurrent) validity was measured by Intraclass Correlation Co-efficient (ICC).

**Table 1. Assessment of content validity.**

| Content validity measure | Numerator | Denominator | Inference |
|---|---|---|---|
| I-CVI | Number of experts who scored item as relevant | Total number of experts | I-CVI values [27] |
| | | | Less than 0.7 = Needs Elimination |
| | | | 0.7–0.79 = Needs Revision |
| | | | Higher than 0.79 = Appropriate |
| S-CVI | | | S–CVI values greater than 0.8 [23–25] is acceptable content validity for the tool. |
| S–CVI (Ave) [Average approach] | The average proportion of items rated as 2 or 3 across the various judges i.e., Sum of I-CVIs | Total number of items (32 items) | |
| Modified Kappa statistic (K) | K = (I-CVI–$P_c$) | (1 –$P_c$) | Modified Kappa values [28] |
| | | | 0.8–1 = almost perfect |
| | | | 0.6–0.8 = substantial |
| | | | 0.4–0.6 = fair |
| | | | 0.2–0.4 = moderate |
| | | | 0–0.2 = light |
| | | | 0 = no agreement |

Pc = Probability of chance agreement.

Probability of chance agreement was first calculated for each item by following formula: $P_c = [N! / A! (N–A)!]^* .5^N$ where N = Number of experts in the panel (8) and A = Number of experts who agree that the item is relevant.

**Table 2. Factor extraction matrix and constructs.**

| SL | | Pattern Matrix[a] | | | |
|---|---|---|---|---|---|
| | | Factor | | | |
| | | 1 | 2 | 3 | 4 |
| | **Construct: Effort–Reward, Organizational Support and Job Security** | | | | |
| ST7 | Inadequate Recognition | .708 | | | |
| ST4 | Lack of Respect | .688 | | | |
| ST5 | Lack of Promotion | .661 | | | |
| ST13 | Lack of Support and Appraisal | .450 | | | |
| ST6 | Lack of Job Security | .337 | | | |
| | **Construct: Job Control and Multiple Demands** | | | | |
| ST3 | Multiple Demands | | .750 | | |
| ST8 | Longer Working Hours | | .720 | | |
| ST2 | Multitasking and Role Ambiguity | | .673 | | |
| ST1 | Deadline Pressure due to heavy work load | | .628 | | |
| ST16 | Difficulty in Work Life Balance | | .627 | | |
| | **Construct: Working Conditions and Training** | | | | |
| ST12 | Uncomfortable Working Conditions | | | -.686 | |
| ST15 | Not trained | | | -.572 | |
| | **Construct: Inter–Personal Relationships** | | | | |
| ST10 | Friction among colleagues | | | | -.692 |
| ST11 | Difficulty in Delegation | | | | -.681 |
| ST14 | Lack of support from colleagues | | | -.349 | -.573 |
| ST9 | Lack of involvement in decision making | | | | -.362 |

Extraction Method: Principal Axis Factoring.

Rotation Method: Oblimin with Kaiser Normalization.[a]

a. Rotation converged in 7 iterations.

**Assessment of construct validity.** Construct validity is the degree to which an instrument measures the trait or theoretical construct that it is intended to measure [33–35]. Construct validity was assessed by Exploratory Factor Analysis (EFA) with the Principal Axis Factoring. The sample size was adequate to conduct EFA since the ratio of the sample size to the number of freely estimated parameters was greater than 10:1. Barlett's Test of Sphericity indicated (significance level $p < 0.05$) that there is a patterned relationship, and Kaiser-Meyer-Olkin Measure of Sampling Adequacy of 0.843 (Table 2) indicated TAWS– 16 is suitable for Factor Analysis (KMO values > 0.7 is good sampling adequacy) [36].

Exploratory Factor Analysis was performed to summarize in a way that relationships and patterns among the domains can easily be understood. The factors that explain the highest proportion of variance, the corresponding variables share is expected to represent the underline constructs. After factor extraction, factors were rotated for better interpretation since unrotated factors are ambiguous. For factor analysis we adopted Direct Oblimin (Oblique Rotation) method [20]. Finally, naming of the factors were done for the final constructs of TAWS– 16.

The data collection was initially planned as face-to-face interview with study subjects but we changed the approach due to COVID– 19 pandemic in Bengaluru city. A web-application of TAWS– 16 was developed, following the complete life cycle of software development. The IT employees were contacted from consenting IT companies in Bengaluru city and web link (https://app.esamiksha.in/stress_assessment) was provided to submit their responses. We

received a total of 356 responses. The application was developed in a manner to administer both TAWS– 16 and DASS– 21 for completing the data collection to assess criterion validity. Data from the application was provided to the investigator as '.csv' file which was converted to MS Excel to check for duplicates, outliers and consistency in responses and coding. Data Analysis was done using SPSS version 25. Data privacy and confidentiality were ensured as it was anonymous and de–linked data collection, with each subject creating his/her own user ID and password. Identifiers like name of the person, phone number or company name or residence details were not collected.

## Results

### Face validity

The average scores (out of 10) were 8.41± 0.3 for all overall face validity and 8.62± 0.39, 8.41 ±0.42, 8.64±0.34 and 7.98±0.65 for level of understanding, unambiguity, clarity and less chance of misinterpretation, respectively.

Out of 16 items in Section-A, 11 items scored ≥ 8 for level of understanding, 13 items scored ≥ 8 for unambiguity, 6 items scored ≥ 8 for less chance of misinterpretation and all the 16 items scored ≥ 8 for clarity. Assessment revealed good face validity of TAWS– 16. Based on face validity assessments, 2 items (Item 3 and Item 9) were slightly modified. Detailed tables are available in supporting information (S1 Table).

### Content validity

I–CVI (Item level): The I–CVI for the items pertaining to work-stress (Section A, denoted as ST) and symptoms suggestive of work-stress (Section B, denoted as SY) varied from 0.375–1 and 0.5–1, respectively.

For work-stress assessment (Section A), four items (ST1, 2, 10 and 16) obtained highest score of 1. Out of 16 items from Section A, 10 items (62%) had I–CVI values > 0.79, which is appropriate. Two items (13%, ST7, 13) needed revision. For assessment of symptoms suggestive of work-stress (Section B), 6 items (SY1, 2, 8, 9, 10 and 11) obtained the highest score of 1. Out of 16 items in Section B, 12 items (75%) had I–CVI values > 0.79 which is appropriate. Four items (23%, SY4, 7, 12 and 16) needed revision. Scale level content validity (S-CVI/Average score) was calculated as 0.829 which is a good score at a scale level. Overall, TAWS-16 demonstrated good content validity. Detailed tables are provided as supporting information (S2 Table).

Item wise modified kappa statistics for work-stress and symptoms suggestive of work-stress, ranged between 0.2–1 and 0.31–1, respectively. For work-stress (Section A): Four items (ST1, 2, 10 and 16) obtained highest score of 1. Out of 16 items, 14 items (87%) had agreement ranged between "Almost Perfect" to "Moderate". Two items (13%, ST12 and 15) had "Fair" agreement. For Symptoms Suggestive of work-stress (Section B), Six items (SY1, 2, 8, 9, 10 and 11) obtained highest score of 1. Out of 16 items in Section B, 15 items (94%) had agreement ranged between "Almost Perfect" to "Moderate". One item (6%, SY13) had "Fair" agreement.

Based on both Content Validity Index (I–CVI and S–CVI) and Modified Kappa Statistics, the TAWS– 16 elicited overall acceptable Content Validity.

### Construct validity

While assessing empirical validity (construct and criterion), majority of the respondents were male (63.2%) with mean age of 33.2 years. Nearly 67% of the respondents were married and 44.1% of the respondents had working experience of 4–7 years with an average of 6.04 years of work experience. Socio-demographic details of total 356 respondents are provided in S1 and S2 Tables.

Table 3. Factor correlation matrix.

**Factor Correlation Matrix[a]**

| Factor | 1 | 2 | 3 | 4 |
|---|---|---|---|---|
| 1 | 1.000 | .099 | -.397 | -.523 |
| 2 | .099 | 1.000 | -.108 | -.137 |
| 3 | -.397 | -.108 | 1.000 | .353 |
| 4 | -.523 | -.137 | .353 | 1.000 |

[a]Extraction Method: Principal Axis Factoring. Rotation Method: Oblimin with Kaiser Normalization.

Barlett's Test of Sphericity indicated (significance level $p < 0.05$) that there is a patterned relationship, and Kaiser-Meyer-Olkin Measure of Sampling Adequacy of 0.843 proved that the study questionnaire is suitable for Factor Analysis [36]. We used "Rotation Sums of Squared Loadings" for the extraction of significant factors. From the Rotated Component Matrix and Pattern Matrix, we extracted 4 factors which explained high proportion of total variance. A factor was retained when it has minimum three variables and item loadings of > 0.32. Since, all the 4 factors fulfilled this criterion, all of them were retained for final construct. Direct Oblimin rotation technique produced symmetrical off–diagonal element (Table 3) and hence found suitable for this. Nearly 61% of the total variance could be explained by the factors identified in our study.

Finally, naming of the factors were done for the final constructs of the study questionnaire (Table 2). We identified 4 dimensions (factors) which measure the construct. 'Effort–Reward, Organizational Support and Job Security' included ST7, 4, 5, 13, 6; 'Job Control and Multiple Demands' included ST3, 8, 2, 1 and 16; 'Working Conditions and Training' included ST12, 15 and 'Inter–personal Relationships' included ST10, 11, 14, 9.

## Criterion validity

Assessment of criterion validity of TAWS– 16 against stress-subscale of DASS– 21, revealed that 292 respondents (82%) 'did not have work-stress' as per both TAWS– 16 and DASS– 21 while 54 respondents (15.2%) who have stress according to TAWS– 16, did not have stress as per DASS– 21 (stress component). The level of agreement was 'moderate' between the two study instruments (Kappa statistic = 0.208) (Table 4). Scores of TAWS– 16 was tested against scores of DASS– 21 (stress component) revealed a Intraclass Correlation Coefficient value of 0.434 (95% CI 0.346–0.515), indicating lower criterion validity between the two.

## Discussion

Validity of TAWS– 16 was tested on an adequate and convenient sample of managerial–supervisory employees from IT sector, as they form a key working population in Bengaluru city. The key strength of the study is testing a much needed, easy to use and interpret work-stress

Table 4. Criterion validity–TAWS-16 v/s stress component of DASS– 21.

| TAWS– 16 | DASS -21 (Stress Component) (n = 356) | | Total (%) |
|---|---|---|---|
| | *Stress Absent (%)* | *Stress Present (%)* | |
| *Stress Absent (%)* | 292 (82) | 1 (0.3) | 293 (82.3) |
| *Stress Present (%)* | 54 (15.2) | 9 (2.5) | 63 (17.7) |
| *Total (%)* | 346 (97.2) | 10 (2.8) | 356 (100) |

Kappa Value = 0.208, P = <0.0001, Percent Agreement: 292+9/356*100 = 84.5%

assessment tool (TAWS– 16) which can increase coverage of work-stress screening in Indian workplaces. Other notable strength is developing an e-application for data collection to increase accuracy and completeness. Despite ongoing COVID– 19 situation, e-application ensured completion of data collection.

TAWS– 16 enquires and combines work-stressor exposure of employees, their coping abilities to exposed stressor and their experience of psychosomatic symptoms, to identify and classify work-related stress. This triad approach is missing in many available work-stress tools and TAWS– 16 bridges this gap. Being available as a weblink, it is easy to administer, anonymous and provides results to employees immediately after assessment. This tool is already used in various industries and is reported to be useful.

During assessment of Face and Content Validity, experts were chosen from fields of occupational health, public health, Information Technology (IT) companies, psychiatry and psychology. We observed that experts from occupational health, public health and IT companies provided similar ratings whereas minor differences were reported from psychology experts. Couple of items scored high by occupational health, public health and IT experts were retained even though they were scored average by a psychologist, owing to their better experience in on-site work stressor management as against a psychologist tending to cases in a clinical set-up. We also modified the wordings of items based on suggestions received from the experts.

Content Validity Index is most widely used to quantify content validity based on relevance and adequacy of each item in the study instrument [37]. Though I–CVI values of few items were low, they represented key work-stressor domains and were rated high by the experts from the occupational health field. Such items were retained with modification and this did not affect the overall scale level content validity index (S–CVI).

Criterion Validity is assessed when one is interested in determining the relationship of scores on a test to a specific highly rated existing standard (gold standard) instrument. Unfortunately, no such instrument specific for work related stress is available in India. In our study, we did not achieve a satisfactory level of criterion validity as there was no appropriate comparison instrument. However, we compared TAWS– 16 with stress section of DASS– 21 [32]. We observed that DASS– 21 lacked constructs specific to work-stress and work stressors, which is the key reason for unsatisfactory results for criterion validity. There is a need for reassessing criterion validity with other instruments from other countries like Occupational Stress Scale [38], Job Description Index (JDI) [39], HSE's Stress Management Standards Indicator Tool [40] and Job Content Questionnaire [41]. In construct validity assessment, the factors in the present study had a clear and precise latent implication allowing easier interpretation. The contents described by the items loading onto each factor in this study were highly coherent except for "working condition" and "training" in Factor 3.

NCDs and NCD risk factors are increasing in working population and harmful work-stress is a potential risk factor for NCDs, mental health and substance use disorders. From the organisational perspective, work-stress can increase burn out, absenteeism and presenteeism which interferes with work productivity. Yet, screening for work-stress is not included during periodical health check-up of employees. This could be due to lack of standard stress assessment tools, lack of priority and doubts regarding its implementation feasibility. However, studies have shown it is feasible to conduct screening for psychological distress and stress [18] during annual medical examination in industries.

TAWS– 16 has scope for integration into existing Periodical Medical Examination (PME) and its inclusion in PME will make PME more comprehensive and help to better understand associations between work-stress levels with the biochemical and metabolic parameters of NCDs and NCD risk factors. It would provide a comprehensive picture of employees' overall health and wellbeing.

TAWS– 16 has the scope to be used either as self–reported or interviewer administered schedule. Though in this assessment, it was used as a self-administered tool as the target population were educated subjects, TAWS– 16 was also used by researchers and occupational physicians covering nearly 2000 employees, where it was interviewer administered. Details cannot be revealed due to non-disclosure agreements.

TAWS– 16 is objective in assessment providing opportunities to monitor changes in stress scores over time and also evaluate effectiveness of stress management and health promotion interventions. As a valid measure, it has both prognostic and monitoring value. It is in-line and contributes towards achieving NCD related Sustainable Development goals in India.

The limitations were lack of valid work-stress tool to test criterion validity. Due to ongoing COVID– 19 situations, data collection was done using digital platform using convenient sampling. This might result in information bias and lack of representativeness. The COVID situation also unfolded newer dimensions to work-stress that warrants further research and interventions.

## Conclusion

In conclusion, the TAWS– 16 has a clear and meaningful structure in accordance with occupational stress theory. All the parameters of validity assessments were satisfactory and within acceptable limits. TAWS– 16 can be used as an instrument to measure work related stress among employees and has scope to be integrated into periodical health check-up of employees in India.

## Supporting information

**S1 Table. Rating of face validity of items of TAWS– 16 by experts.**
(DOCX)

**S2 Table. Rating and calculation of I-CVI and S-CVI of work related factors and symptoms suggestive of work stress for TAWS– 16.**
(DOCX)

## Acknowledgments

The authors would like to express their sincere gratitude to AKA Labs team for their support and development of the web-based application for TAWS– 16. Most significantly, our sincere thanks to the domain experts and IT professionals for their cooperation and participation in the study.

## Author Contributions

**Conceptualization:** Runalika Roy, Gautham Melur Sukumar, Gururaj Gopalakrishna.

**Data curation:** Gautham Melur Sukumar, Mariamma Philip.

**Formal analysis:** Runalika Roy, Gautham Melur Sukumar, Mariamma Philip.

**Investigation:** Gautham Melur Sukumar.

**Methodology:** Runalika Roy, Gautham Melur Sukumar, Mariamma Philip.

**Project administration:** Gautham Melur Sukumar.

**Resources:** Gautham Melur Sukumar.

**Supervision:** Gautham Melur Sukumar, Gururaj Gopalakrishna.

**Validation:** Runalika Roy, Gautham Melur Sukumar.

**Visualization:** Gautham Melur Sukumar.

**Writing – original draft:** Runalika Roy, Gautham Melur Sukumar.

**Writing – review & editing:** Runalika Roy, Gautham Melur Sukumar, Gururaj Gopalakrishna.

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
