## [Decision Letter · Decision Letter 0]

10 Nov 2022

PONE-D-22-20570Face, content, criterion and construct validity assessment of a newly developed tool to assess and classify work–related stress (TAWS – 16)PLOS ONE

Dear Dr.Gautham Melur Sukumar,

Thank you for submitting your manuscript to PLOS ONE. After careful consideration, we feel that it has merit but does not fully meet PLOS ONE’s publication criteria as it currently stands. Therefore, we invite you to submit a revised version of the manuscript that addresses the points raised during the review process.

We look forward to receiving your revised manuscript.

Kind regards,

Sebsibe Tadesse, PhD

Academic Editor

PLOS ONE

Reviewers' comments:

Reviewer's Responses to Questions

**Comments to the Author**

1. Is the manuscript technically sound, and do the data support the conclusions?

Reviewer #1: Yes

2. Has the statistical analysis been performed appropriately and rigorously? 

Reviewer #1: Yes

3. Have the authors made all data underlying the findings in their manuscript fully available?

Reviewer #1: Yes

4. Is the manuscript presented in an intelligible fashion and written in standard English?

Reviewer #1: No

5. Review Comments to the Author

Reviewer #1: Thank you for the research, I have comments and questions as follow:

1- please report the CVR for content validity.

2- for data collection(items) what do you do? interview ? in which method? qualitative research in which method? is it mixed method? this part is ambiguous

3- likelihood is better than PCA please explain why PCA selected

4- Table 2 can be eliminated and the data report in the text

Good Luck

6. PLOS authors have the option to publish the peer review history of their article (what does this mean?). If published, this will include your full peer review and any attached files.

Reviewer #1: **Yes: **Dr Malahat Akbarfahimi

---

## [Author Response · Author response to Decision Letter 0]

9 Dec 2022

Journal Requirements comments (1, 2 and 3 are addressed).

Reviewer 1

A. please report the CVR for content validity

CVI and Modified Kappa are reported as measure of content validity in our study as CVI and Kappa are reported as better and commonly used measure of content validity [1]. We reported Modified Kappa to eliminate chance agreement

Based on the study by Ayre and Scally [2], the critical value of CVR for 8 experts would be 0.75. Our experts ranged from diverse occupational health backgrounds (occupational health experts, public health experts, psychiatrists, psychologists, IT professionals) due to which CVR critical value was difficult to achieve for items which were otherwise found appropriate with CVI and Modified Kappa. Hence, we did not use CVR as a measure of content validity in our analysis.

1. Wynd CA, Schmidt B, Schaefer MA. Two quantitative approaches for estimating content validity. Vol. 25, Western Journal of Nursing Research. 2003. p. 508–18. 

2. Ayre C, Scally A. Critical Values for Lawshe’s Content Validity Ratio. Meas Eval Couns Dev. 2013 Dec 13;47:79–86.

B. For data collection(items) what do you do? interview? in which method? qualitative research in which method? is it mixed method? this part is ambiguous.

For face validity and content validity – Hard copy of the study instrument was personally handed over to the experts and context was explained to them. For experts staying outside Bengaluru city, the form was emailed, and context was explained over telephone. The experts filled out the form and provided it back to the investigator

For assessment of criterion and construct validity, the investigator made a phone call to random IT employees and sought verbal consent for participation. For employees who were willing to participate, a web-link to TAWS-16 was emailed to them. The participating employees created a login and password and provided informed consent before filling out the questionnaire in email (self-reporting).

No qualitative research techniques were employed in this study. 

Same is mentioned in the materials and methods

C. likelihood is better than PCA please explain why PCA selected

We used EFA and likelihood for construct validity analysis. The same is mentioned in the materials and methods.

PCA was not done.

D. Table 2 can be eliminated and the data report in the text

The previous one has been eliminated and data reported in the text and renamed

---

## [Editor Report · Decision Letter 1]

22 Dec 2022

Face, content, criterion and construct validity assessment of a newly developed tool to assess and classify work–related stress (TAWS – 16)

PONE-D-22-20570R1

Dear Dr. Gautham Melur Sukumar,

We’re pleased to inform you that your manuscript has been judged scientifically suitable for publication and will be formally accepted for publication once it meets all outstanding technical requirements.

Kind regards,

Sebsibe Tadesse, PhD

Academic Editor

PLOS ONE

---

## [Editor Report · Acceptance letter]

29 Dec 2022

PONE-D-22-20570R1 

Face, content, criterion and construct validity assessment of a newly developed tool to assess and classify work–related stress (TAWS – 16) 

Dear Dr. Sukumar:

I'm pleased to inform you that your manuscript has been deemed suitable for publication in PLOS ONE. Congratulations! Your manuscript is now with our production department. 

Kind regards, 

on behalf of

Dr. Sebsibe Tadesse 

Academic Editor

PLOS ONE